# Pharmacists’ Knowledge of Factors Associated with Dementia: The A-to-Z Dementia Knowledge List

**DOI:** 10.3390/ijerph18199934

**Published:** 2021-09-22

**Authors:** Hernán Ramos, Lucrecia Moreno, María Gil, Gemma García-Lluch, José Sendra-Lillo, Mónica Alacreu

**Affiliations:** 1Cátedra DeCo MICOF-CEU UCH, Universidad Cardenal Herrera-CEU, 46115 Valencia, Spain; ramgarher@alumnos.uchceu.es (H.R.); lmoreno@uchceu.es (L.M.); maria.gil@micof.es (M.G.); gemma.garcia2@alumnos.uchceu.es (G.G.-L.); jsendralillo@gmail.com (J.S.-L.); 2Muy Ilustre Colegio Oficial de Farmacéuticos, 46003 Valencia, Spain; 3Department of Pharmacy, Universidad Cardenal Herrera-CEU, CEU Universities, 46115 Valencia, Spain; 4Embedded Systems and Artificial Intelligence Group, Universidad Cardenal Herrera-CEU, CEU Universities, 46115 Valencia, Spain

**Keywords:** knowledge, dementia, pharmacist, risk factors, protective factors, pharmaceutical drugs, cognitive impairment

## Abstract

Dementia is a neurodegenerative disease with no cure that can begin up to 20 years before its diagnosis. A key priority in patients with dementia is the identification of early modifiable factors that can slow the progression of the disease. Community pharmacies are suitable points for cognitive-impairment screening because of their proximity to patients. Therefore, the continuous training of professionals working in pharmacies directly impacts the public health of the population. The main purpose of this study was to assess community pharmacists’ knowledge of dementia-related factors. Thus, we conducted a cross-sectional study of 361 pharmacists via an online questionnaire that quizzed their knowledge of a list of dementia-related factors, which we later arranged into the A-to-Z Dementia Knowledge List. We found that younger participants had a better knowledge of risk factors associated with dementia. The risk factors most often identified were a family history of dementia followed by social isolation. More than 40% of the respondents did not identify herpes labialis, sleep more than 9 h per day, and poor hearing as risk factors. A higher percentage of respondents were better able to identify protective factors than risk factors. The least known protective factors were internet use, avoidance of pollution, and the use of anti-inflammatory drugs. Pharmacists’ knowledge of dementia-related factors should be renewed with the aim of enhancing their unique placement to easily implement cognitive-impairment screening.

## 1. Introduction

Dementia is a currently uncurable, long-term neurodegenerative disease that can begin up to 20 years before its diagnosis. It is a major global health problem, with more than 50 million people estimated to be living with the disease worldwide, a number that will increase to 152 million by 2050 [1]. According to the Statistical Office of the European Union (Eurostat), in 2017, Spain had the third highest number of deaths caused by dementia in Europe, after United Kingdom and Germany. In that year, 36,185 people died because of this pathology, representing 8.5% of all deaths in Spain [2].

The scientific literature indicates that dementia is a multifactorial disease whose development has no single cause, although it seems that age at the time of diagnosis is the most influential factor [3]. Therefore, dementia risk factors should be detected and modified at the stages in which they are most determinant for the development of the pathology.

According to certain publications, years of education are highly important, since it has been estimated a 7% dementia reduction when less education is avoided [4]. In contrast, people with more years of education seem to have less risk of developing dementia, even if they have well-known dementia genetic risk factors such as APOE ε4 [5]. Regarding midlife, hearing loss appears to be the most influential factor, accounting for 8% of the dementia modifiable risk, followed by traumatic brain injury (3%), hypertension (2%), alcohol (1%) and obesity (1%) [3,4]. Not included among those factors, but probably important as well, may be the presence of hypercholesterolemia [5,6,7,8,9] and the exposure to heavy metals or to certain mycotoxins, produced by molds or cyanobacteria [10,11]. Finally, smoking (5% of the total modifiable dementia risk), depression (4%), social isolation (4%), physical inactivity (2%), air pollution (2%) and diabetes (1%) may contribute to an increased risk of dementia at later life. Consequently, adding the above factors, up to 40% of the risk of dementia can be avoided [4].

Thus, a key priority in patients with dementia is the identification of early modifiable factors that will decrease the risk, delay the onset, or slow the progression of the disease. The research by Livingston et al. highlighted the value of addressing multiple dementia risk factors as a strategy to protect brain health and promote overall well-being and functioning (http://wwfingers.com) (accessed on 20 April 2021). Indeed, some risk factors, mainly those present in later life, can constitute an adequate prevention strategy in themselves [4]. Thus, early interventions can only be effective and implemented if the population that stands to benefit from them can be identified.

In this context, projects that aim to prevent dementia by disseminating information about the risks and protective factors for the disease and which promote the early detection of individuals with dementia are of particular importance [12,13,14]. To achieve that goal, many countries support the use of dementia prevention guidelines by training healthcare personnel. This work is being supported at the country level through the adoption of the Global Action Plan on the Public Health Response to Dementia 2017–2025; the Comprehensive Mental Health Action Plan 2013–2020; and the Global Action Plan for the Prevention and Control of Noncommunicable Diseases 2013–2020 [3].

Among other entities, the Lancet Commission, Alzheimer’s Disease International (ADI), and the World Health Organization (WHO) have published different risk and protective factors related to dementia [3,15]. Nevertheless, in recent years, variables such as subjective memory complaints (SMCs), the use of some medications, patient knowledge of risk factors, internet usage, and certain infections are also gaining importance [16,17,18]. Therefore, it is vital to detect weaknesses in professional healthcare knowledge and to enhance a lifelong learning among these professionals, especially emphasising the lesser-known aspects of the pathology.

Community pharmacies stand out from among the different healthcare centres because of their proximity to patients. A good example of the importance of this factor is the present context caused by the COVID-19 pandemic; community pharmacies remained open during the pandemic lock-down and served as unique first-line healthcare centres which opened for face-to-face consultations for non-urgent pathologies when hospitals and outpatient clinics were otherwise overwhelmed [16].

Given that community pharmacies are the nearest and most frequented healthcare centres visited by patients, emphasising education about dementia among pharmacists would help to detect cognitive decline earlier and raise patient awareness of the modifiable lifestyle factors they could address to help improve their brain health. Some studies have also highlighted the huge role that pharmacists can play in terms of improving social health when they are confident in their knowledge, especially in reference to mental health [16].

Therefore, as shown in previous studies [12], community pharmacies are a suitable point for cognitive impairment (CI) screening and the interprofessional collaboration between pharmacists and physicians improves dementia diagnosis rates [14]. In the same line, community pharmacies represent a point of quality health information for many patients and so, continuous training of these professionals can directly impact the public health of the population. Thus, the main purposes of this scientific study were to (1) elaborate an A-to-Z of Dementia Knowledge List containing known associated factors, (2) assess the knowledge of managing community pharmacists regarding dementia and its related factors, and (3) to elaborate supporting material to improve the main gaps of knowledge identified in this area of dementia result of a bibliographic review.

## 2. Materials and Methods

(A) Bibliographic review

We wanted to search for new modifiable risk or protective factors and to assess those already established by renowned medical journals such as The Lancet [4] or prestigious institutions including the WHO [3] or ADI [1]. Thus, we conducted a thorough review of the PubMed and Web of Science databases, applying the key word and inclusion and exclusion criteria shown in Table 1. We focused on the modifiable factors least known by pharmacists and least mentioned by our colleagues.

(B) Questionnaire and data collection

We created a survey to ascertain pharmacists’ knowledge of the dementia risk and protective factors identified in the bibliographic review, the A-to-Z Dementia Knowledge List. The survey was developed and refined by the alliance between the Official Illustrious College of Pharmacists in Valencia (Muy Ilustre Colegio Oficial de Farmacéuticos de Valencia, or MICOF) and Cardenal Herrera University (UCH) for the detection of cognitive impairment (DeCo in its Spanish portmanteau (https://www.uchceu.es/catedra/deco) (accessed on 10 July 2021) study team (who were all qualified pharmacists and research scientists) based on their experience and expertise. We then internally validated the questionnaire, which was piloted among the pharmacy degree students at the university. A group of neurologists supervised and approved the final version of the survey, and its full version can be found in the Appendix A. which was piloted among the pharmacy degree students at the university. A group of neurologists supervised and approved the final version of the survey, and its full version can be found in the Appendix A.

The questionnaire was a 40-item form with 4 questions about the profile of the respondents (sex, age, years of experience, and close relatives with dementia) and 36 statements about factors associated with dementia. Within those factors, 16 questions were about risk factors, 14 concerned protective factors, and 5 questions related to factors not associated with dementia. The respondents had to decide whether the situation described was a risk factor for developing dementia, a protective factor, or not associated with developing dementia, or they could decide not to answer the question if they did not know the answer.

(C) Participants

A cross-sectional study was conducted during February 2021. All pharmacists belonging to the MICOF Valencia were invited to participate via an online questionnaire using Microsoft Forms to collect the data. A total of 4881 collegiate members received the survey with questions about the A-to-Z Dementia Knowledge List by email, which they could also forward to their colleagues. The link was open for a month and a reminder email was sent out before the survey closed. The questionnaire responses were anonymised.

(D) Statistical analysis

Based on the classical statistical formula (*n* = [(Z_1−α/2_)^2^ σ^2^]/E^2^) [19] for the calculation of the required sample size in a study on the estimation of a mean score, with 95% confidence (i.e., α = 0.05; Z_0.975_ = 1.96), in a questionnaire with an estimated standard deviation of 4 points (σ ≈ 4) and an error E = 0.5 (i.e., precision 2E of 1 point), 246 participants were needed. According to our estimations, a 5% of response rate was needed to achieve our objective.

The respondents’ responses were automatically stored in an Excel spreadsheet, and the three records mentioned above were eliminated. To improve the subsequent description of the results, the participant ages were categorised into 4 groups and the years of experience variable was categorized into 5 groups. Likewise, a dichotomous variable was added for each of the 36 factors evaluated in the survey, whereby 1 point was awarded for a correct answer and 0 points for an incorrect answer for that factor. The database was expanded to add these new data as well as 4 more variables: the total number of correctly identified (1) risk factors (with a maximum score of 16 points per respondent), (2) protective factors (with maximum score of 15 points per respondent), (3) non-associated factors (with maximum score of 5 points per respondent), and overall, correctly identified factors (with maximum score of 36 points per respondent).

We wanted to determine whether the mean number of correct answers for dementia risk factors, protective factors, and non-associated factors was significantly different for the 3 categories of variables determining the respondent profiles. Therefore, we used the Levene test to analyse the variability of the scores; ANOVA to contrast more than 2 mean scores, Tukey test for multiple comparisons of mean scores, and Student *t*-test for comparison of two means in independent samples. Likewise, we also calculated the percentage of respondents who considered each of the 36 survey factors a risk factor, protective factor, or non-associated factor for developing dementia, or were an unknown factor, arranging these in descending order.

The technical statistical treatment was carried out using R advanced statistics software, and both the protection of the data and its security were guaranteed. The information was treated with confidentiality and in a lawful way and for the purpose the respondents were informed of, in compliance with the General European Data Protection Regulation (GDPR) and Organic Law 3/2018 on Protection of Personal Data and the guarantee of digital rights. In addition, this work complied with the basic principles of the Declaration of Helsinki.

(E) Ethical approval

The participation was anonymous. Before completing the initial scale, the volunteer respondents were assured of the data confidentiality and informed that sending a completed online survey was considered consent for the inclusion of the non-identified data in our reports. This study was approved by the Research Ethics Committee of the *Universidad Cardenal Herrera CEU* (UCH-CEU; approval No. CEI21/054).

## 3. Results

Due to the idiosyncratic characteristics of this article and to facilitate data comprehension, the results were divided into 3 points.

### 3.1. Bibliographic Review of Factors Related to Alzheimer Disease: The A-to-Z Dementia Knowledge List

To make the dementia risk factors easier to remember and memorise, first we elaborated the A-to-Z Dementia Knowledge List (Table 2). There was clear evidence of dementia risk factors from some of the citations (Livingston et al., 2020; WHO; ADI), we also included some less well-known factors that were also important such as SMCs, knowledge, pharmaceutical drugs, bacterial or virus infections, and internet use.

### 3.2. New Risk Factors Added to Those Already Cited by Recognised Organisations

Based on the new risk factors, a literature review was carried out (Table 3), identifying and selecting the most relevant articles for each risk factor in the last 3 years. This resulted in an additional five risk factors being added to those already cited by recognised organisations. Subjective memory complaints such as alertness; knowledge about dementia as a protective factor, internet use as a cognitive stimulant. Additionally, pharmaceutical drugs as treatments or as a risk factor (depending on their effects), and finally, bacterial and virus infections. Each risk factor has been discussed subsequently.

### 3.3. Survey Results

From the 4881 members of the College of Pharmacy in Valencia who were initially contacted, 364 people accessed the survey. Of these, two declined to complete the questionnaire and one was underage. Thus, 361 pharmacists completed the survey, corresponding to 7.4% of the total number of collegiate members; 258 of these pharmacists were women (71.5%) and 103 were men (28.5%). Table 4 shows 3 variables that define the respondents’ profiles: age, categorised as four groups; years of experience (5 groups); and close relatives with dementia. These variables were compared to each set of questionnaire items, distinguishing risk factors (16 items), protective factors (15 items), and factors not associated with developing dementia (5 items). The mean number of correct responses and standard deviation were calculated to summarise each profile variable.

The uncategorised variables age and years of work experience were strongly positively correlated (Pearson correlation test: *p*-value < 0.001). Of note, the youngest participants (aged 20–30 years) got a higher mean number of correct answers about the risk factor items than those aged 60–75 years old (Tukey multiple comparisons test: *p*-value < 0.05). Similarly, participants with less work experience (0–10 years) achieved significantly more correct answers than those with 10–30 or 40–50 years of work experience (Multiple Tukey comparisons test: *p*-value < 0.05 and *p*-value < 0.01, respectively).

Regarding the protective factors, participants with less work experience (0–10 years) obtained a significantly higher mean number of correct answers than those with 10–20 years of work experience (Tukey multiple comparisons test: *p*-value < 0.05). There were no significant differences between participants who had a relative or close friend with dementia and those who did not in terms of the mean number of correct answers in any of the 3 groups of items (risk factors, protective factors, or factors with no association).

Table 5 details the 36 statements on the questionnaire and the distribution of the responses we received for each question. As shown, there were four possible answers to each statement as follows: risk factor for developing dementia, a protective factor against dementia, no association with dementia, or no answer (if respondent did not know the answer or did not want to answer). Note that, in Table 5, the correct answer for each statement is marked in bold.

In Figure 1, we organised the information shown for the 36 items of the questionnaire in Table 5 into three different graphs. The graph on the left shows the 16 items whose correct response was a risk factor; the central graph comprises the 15 items whose correct response was a protective factor; and the graph on the right is made up of the 5 items whose correct response was that there is no association of the factor with the development of dementia. The items are named in abbreviated form on the *x*-axis along with the questionnaire item number. In addition, the items on each graph are shown in descending order with respect to the percentage of respondents who got the correct answer.

The 16 items included in the graph on the left correspond to some of the main risk factors for developing dementia. As shown, practically all the respondents identified a family history of dementia as a risk factor for developing dementia (94.7%), followed by social isolation (88.1%). More than 40% of the respondents did not identify the last 6 items shown on the graph on the left (including herpes labialis, sleeping more than 9 h a day, and poor hearing) as risk factors, with many participants answering that these factors were not associated with the development of dementia or that they did not know the answer. It was surprising that some respondents identified the use of anticholinergic drugs or sleeping more than 9 h a day as a protective factor against developing dementia.

The central graph shows the 15 questionnaire items whose correct response was a protective factor against dementia. Thus, these protective factors should be recommended as a means to help patients avoid suffering from dementia, either among those who have not yet started experiencing CI or as a shock-treatment in patients who have already entered the CI phase. As shown, more respondents correctly identified the protective factors than the risk factors. For example, almost all the respondents correctly identified cognitive stimulation and daily reading as protective factors against developing dementia (96.7% and 96.4%, respectively). The respondents had greater difficulty in correctly answering the last 5 items shown on the central graph. Of note, 34% and 42.4% of those surveyed thought that maintaining low blood pressure and having a low-high level of education were not associated with developing dementia, respectively. The least known protective factors were internet use, living in a rural area, and use of anti-inflammatory drugs.

Finally, the graph on the right shows the responses to the 5 items whose correct answer was that there was no scientific association with the development of dementia. Interestingly, 35.2%, 76.7% and 88.9% of those surveyed believed that maintaining bodily hygiene, listening to music, or doing crafts was a protective factor against developing dementia, respectively. Thus, it is important that the health personnel designated to make recommendations to patients after screening know that it is playing music rather than listening to music that produces cognitive stimulation and, consequently, a protective effect.

### 3.4. The A-to-Z Dementia Knowledge List Supporting Material

To better understand the factors associated with dementia and help fill in the main gaps in knowledge identified in Table 2, we reorganised the information according to the different stages of dementia and their influence on cognitive decline (Figure 2). As shown, we separated the information according to three main levels of dementia, differentiating the asymptomatic stage, period of MCI, and period in which dementia is evident. In addition, we showed several factors related to each phase, distinguishing several that can influence patients’ evolution to cognitive decline, depending on the disease stage. We chose to depict this as a person wearing a backpack and using an umbrella because it is raining.

The person represents *non-modifiable factors* because *age*, *sex* and *genetics* are all characteristics that cannot be changed. The backpack refers to *individual cognitive reserve* since *educational level* or the *job* a person does are circumstantial factors carried throughout life and are hard to change. The umbrella represents potential *protective factors* like heathy habits (*exercise* and *nutrition*) or cognitive stimulation (solving a *quiz,* surfing on the *internet, reading, meeting friends*, or *playing music)*. Finally, the rain represents *risk factors* such as diseases (*hypertension, insulin resistance, lipid profile alterations, brain injuries, hearing loss, obesity, viral* and *bacterial infections*), environmental exposure to pollution (*zip code*), use of certain *pharmaceuticals* like anticholinergic drugs or benzodiazepines and lastly, *toxic habits* (smoking and alcohol consumption) and environmental toxics (mycotoxins and mercury).

Figure 3 shows a visual document designed to show the main modifiable and circumstantial risk factors for developing dementia and the main protective recommendations for after patient screenings. The two main objectives of this poster were to serve as a reminder reference document for all health personnel involved in dementia screening and second, to promote its dissemination among the general population at risk of belonging to the target population.

## 4. Discussion

After a bibliographic search, which includes the extensive bibliographic revision about dementia risk factors made by The Lancet Commission, actualised on the past year, we have studied and evaluated other factors that we consider important as well.

One of the main contributions of this work was the elaboration of the A-to-Z Dementia Knowledge List to facilitate the task of memorising and remembering dementia risk factors. There is clear evidence for the usefulness of these 21 factors [1,2,3]. In addition, we decided to include five more significant factors (lifestyle factors, pharmaceuticals, SMCs, virus and other infections, and internet use) with the aim to build an alphabet.

The Finnish Geriatric Intervention Study to Prevent Cognitive Impairment and Disability (FINGER) trial showed that an early lifestyle intervention can benefit cognition in elderly people with an elevated risk of dementia [13]. Together with cognitive stimulation activities, minimising risk factors early in midlife can diminish the risk of cognitive decline and delay the development of dementia in later life [20]. Nevertheless, this knowledge will be ineffective unless individuals apply these interventions.

There is a general need for policies to promote active and healthy ageing by engaging, empowering, and motivating patients [21]. In this respect, it has been suggested that a behavioural intervention by means of collaborative care, enhanced patient–health professional interactions, and social support promotes self-efficacy and, consequently, improves patient adherence [22]. Therefore, supporting patients during the implementation of a healthier lifestyle makes them involved in the decision process and helps them to take responsibility for their health [18].

Regarding pharmaceutical drugs, the association between anticholinergic drugs and cognitive decline is well-known, especially at high doses [23,24]. Nevertheless, some studies have failed to find this connection [25]. Similarly, although some authors did not find an association between the use of benzodiazepines and greater cognitive decline [23,26], others did find this relationship [27] as well as a link with a higher risk of adverse events such as falls [26]. Moreover, some authors have linked non-steroidal anti-inflammatory drugs to a decreased risk of AD mortality [28], although clinical trials in this area were unsuccessful. Again, the long-term use of disease modifying anti-rheumatic drugs such as methotrexate also appear to lower the risk of developing dementia [29].

SMCs are also a good predictor of dementia risk, as shown by numerous studies [30,31,32,33] because they are one of the first signs of the loss of alertness. SMCs are an early marker of future cognitive decline and can be incorporated into the diagnosis of mild cognitive impairment (MCI) and neurodegenerative dementias (Table 3). Firstly, older people with SMCs are twice as likely to develop dementia as those without SMCs [32]. Thus, prospective studies have shown that preclinical AD patients with SMC had a 62% risk of progression from MCI to dementia within 3 years [33]. In fact, previous studies have shown that SMC was the variable with the highest discriminatory power in screening for CIs [12] and it has been associated with a greater volume of white matter hyperintensity [34]. In terms of the patient knowledge factor, given the multifactorial aetiology of dementia and Alzheimer’s disease (AD), multidomain interventions that simultaneously target several risk factors and mechanisms might be necessary for an optimal preventive effect.

Regarding lipid profile, there is no clear consensus, although statin treatment has been shown to reduce the risk of dementia in cohort studies [35,36,37,38]. Evidence suggests that high total cholesterol levels in mid-life increased the risk of AD in later life [6]. Moreover, high plasma HDL cholesterol has been observationally associated with an increased risk of dementia and AD [8]. However, a study with 19 years of follow-up found that HDL cholesterol levels in mid-life are inversely associated with MCI and dementia in late life [7]. In addition, LDL cholesterol measured during mid-life was modestly associated with an increased risk of dementia 10 years later [9]. The lack of strong evidence and the existing differences could be explained genetically, as the APOE ε4 genotype impacts differently on the relevance of hypercholesterolaemia in dementia compared to APOE ε4 non-carriers [5]. However, based on recent retrospective studies, an adequate lipid profile is likely to be an indicator of cognitively healthy aging in most of the population.

Furthermore, we would like to highlight the hypothesis that infections are related to dementia, which has recently been gaining attention and is an ongoing field of research (Table 3). Most pharmacists were unaware that herpes labialis is a risk factor for dementia because it is an emerging factor. In 1991, the presence of herpes simplex virus type 1 (HSV-1) was reported in the brains of AD patients [39], but this work was dismissed by most of the scientific community. However, a recent prospective study of 30,000 participants in Taiwan found that patients diagnosed with HSV-1 and herpes simplex virus type 2 (HSV-2) had a 2–2.5-fold higher risk of developing any type of dementia [40]. Consistent with these findings is a recent systematic review [41], and some researchers have found a significant increase in human herpesvirus 6A (HHV-6A) and human herpesvirus 7 (HHV-7) in AD patients compared to controls [42]. Moreover, other microbes have also been associated with AD including *Chlamydia pneumoniae* [43], *Borrelia burgdorferi* [44], and *Porphyromonas gingivalis* [45]. Thus, these findings have opened a new field of study and highlight the importance of treating certain infections to possibly reduce the risk of dementia.

Use of the internet also seems to play a protective role in dementias. Specifically, a randomised study found that surfing the internet had a protective role against CI [46]. Moreover, a study in Brazil also showed a significant association between continued internet use and higher cognitive performance [47]. In this sense, other studies found that internet use was associated with less CI in neuropsychological test [48]. However, the use of digital technologies may provide less benefit to people with dementia than people with MCI, which could reinforce the importance of this factor in the early stages of dementia [49]. This is a new field of study, but all activities that influence cognitive stimulation may help to slow the disease.

In terms of our secondary objectives, this paper presents original results and showcases the A-to-Z Dementia Knowledge List of known factors associated with dementia. This is also the first published work to assess the knowledge of community pharmacists about dementia and its modifiable risk-related factors. Although most of the survey participants were women (71.5% vs. 28.5%), this was proportional to the MICOF membership in Valencia, with 3378 female members and 1,503 male members (69.2% vs. 30.8%). Our most notable finding was that younger pharmacist and those with fewer years of work experience scored better on the survey. This may be because of their more recent university education given that many risks and protective factors related with dementia have been recently discovered. However, having close relatives with dementia did not significantly improve the participants’ mean scores for risk factors (10.2 vs. 9.9), protective factors (10.9 vs. 10.8), or non-associated factors (2.0 vs. 2.1). Of note, that knowledge comes with scientific literacy rather than experience.

A family history of dementia was easily identified as one of the main risk factors for developing dementia. Although family history is not a required criterion, research shows that individuals with a relative with dementia are more likely to develop the disease themselves [50]. Additionally, social isolation is a risk factor for dementia, but can also occur as part of the disease prodrome [51]. In contrast, an association between sleep duration and the risk of MCI or dementia has also been reported, with higher risk being associated with sleeping less than 5 h a day or more than 10 h a day [52]. Interestingly, most of the survey participants mistakenly identified sleeping more than 9 h a day as a protective factor, probably because good rest tends to be associated with a better health status.

As shown in the citations in Table 3 and Figure 1, being aware of dementia risk factors can act as a factor that can prevent or delay dementia. Similarly, correct knowledge of how drugs can impact cognition in elderly people may act as a protective factor or, at least, helps not add to the risk of developing a CI. However, although we can improve lifestyles to try to avoid chronic pathologies or to better control them when they do appear, the availability of suitable therapeutic approaches is also vital. Thus, when possible, the avoidance of benzodiazepines and drugs with an anticholinergic burden should be a key priority in elderly patients (Table 3). When we asked pharmacists about anticholinergic drugs, only half of them where aware that anticholinergic burden is a dementia risk factor. This could be because a wide variety of drugs have anticholinergic properties, and many scales are available to calculate and classify the anticholinergic burden. Conversely, when we analysed pharmacists’ knowledge about dementia association with benzodiazepine use, we found that most responders answered correctly (74%). One of the main reasons for this may be the short-term amnesia this pharmaceutical group causes, as well as the fact that it is frequently prescribed to elderly people in Spain, thereby perhaps increasing pharmacists’ knowledge and awareness in this relationship.

It is well known that cognitive stimulation has a beneficial effect on the development of dementia. In this regard, while the participants clearly linked daily reading with improved cognitive function, they did not identify internet use or speaking multiple languages as activities that promoted the maintenance of cognition. In contrast, the level of education acquired in the first stage of life could be considered a modifiable factor. However, because it is difficult to change, it should be regarded as a circumstantial factor. Education contributes to the cognitive reserve people will carry with them into the future, which may require varying degrees of cognition [53].

The maintenance of low blood pressure also stood out among the other protective factors that were not often correctly identified by the surveyees. This might be because it would have been easier to identify hypertension as a risk factor rather than blood pressure control as a protective factor. Similarly, we asked about living in a rural area because of the associated reduction in pollution levels given that emerging evidence suggests that exposure to high levels of airborne pollutants is associated with an increased risk of dementia [10,54]. We would also like to highlight the items referring to factors not scientifically related to the development of dementia, such as listening to music. We wanted to discern between playing an instrument and listening to music because only playing music has been associated with maintaining cognition and brain health [55].

A previous study using the *Alzheimer’s Disease Knowledge Scale* (ADKS) showed that Spanish healthcare professionals have a good knowledge of the pathology [56]. The ADKS assesses 7 areas of AD: impact, risk factors, course, diagnosis, caregiving, treatment, and symptoms, with the lowest scores being obtained for the risk factors (mental exercise, age, cholesterol, drugs, blood pressure, and genetic). In addition, many professionals were unaware of the influence that high blood pressure or high levels of blood lipids during midlife may have on the development of dementia in old age [57]. Furthermore, a survey in the Maltese Islands using the ADKS and *Alzheimer’s Disease Pharmacotherapy Measure* (ADPM) showed that community pharmacists had an inadequate knowledge of dementia risk factors, caregiving issues, and pharmacological management [58]. This contrasts with the work by Alacreu et al. (2019) in Spain [56], perhaps because of education differences between the two countries. Nevertheless, these findings suggest new training strategies should be developed and implemented to improve the knowledge of medical and pharmaceutical professionals in the field of dementia risk factors.

Finally, we used our results to elaborate supporting educational material, including advertising posters and short explainer videos, with a view to improving community pharmacists’ knowledge of dementia and its associated factors. Pharmacies are usually the first point of contact in the healthcare system, and they have the potential to provide information and life-sustaining support for patients [58]. Our material could also be useful to aid and promote the role community pharmacists already play in health education. For example, the main objective of the graphics shown in Figure 2 and Figure 3 was to train people about dementia and to serve as a reference for learning and as a reminder of factors related to dementia.

As shown in Figure 2, susceptibility to dementia depends on patients’ intrinsic factors, their travelling bag, the strength of their umbrella, and intensity of the rain. Thus, certain alerts such as *subjective memory complaints*, *difficulty in performing tasks*, *behavioural disorders*, or *depression* may appear in patients that could warn us about their dangerous path leading towards dementia. These factors may lead the patient into the dementia stage, starting with the CI phase. Thus, factors like healthy habits, cognitive stimulation, and pharmaceuticals may improve the CI, or at least, could extend the path. This figure serves as a useful metaphor in patient education and as a reminder of the factors related to dementia, which can be distributed to the members of the MICOF.

Importantly, this work had certain limitations. Firstly, it was conducted in one province and thus, the results may not be generalisable to the rest of Spain. Secondly, only 7.4% of the pharmacists we contacted responded to the survey, although this was in line with data obtained in other research that also invited health professionals to participate in an online survey [56,59]. Thirdly, some of the items of our questionnaire may have been misunderstood. For example, in item number 28 when we mentioned manual work, we wanted to refer to activities that do not involve cognitive stimulation, while some manual activities, such as artistic ones, can entail such stimulation. Finally, given that the survey was conducted online, some surveys may have looked up answers while responding.

Further research will be needed to validate the A-to-Z Dementia Knowledge List and to extend its use. Moreover, our results are relevant to future work because these kinds of surveys can be used for the early detection of dementia in the general population. All the educational items elaborated in this work can be widely distributed by pharmacists and other health professionals to contribute to health education and to detect dementia as soon as possible.

## 5. Conclusions

This study identified the factors most commonly documented in the scientific literature as being related to dementia. CI screening requires collaborative teamwork by all healthcare providers, including community pharmacists. Knowledge gaps about these factors in pharmacists should be addressed to enhance their abilities to successfully screen for CI. Moreover, our questionnaire results have shown that up-to-date knowledge is more effective than years of experience. This lack of information about some factors reinforces the importance of developing an A-to-Z Dementia Knowledge List that includes lesser-known risk factors. Finally, the role of community pharmacists as health educators highlights the need to continuously update their scientific knowledge.

## Figures and Tables

**Figure 1 ijerph-18-09934-f001:**
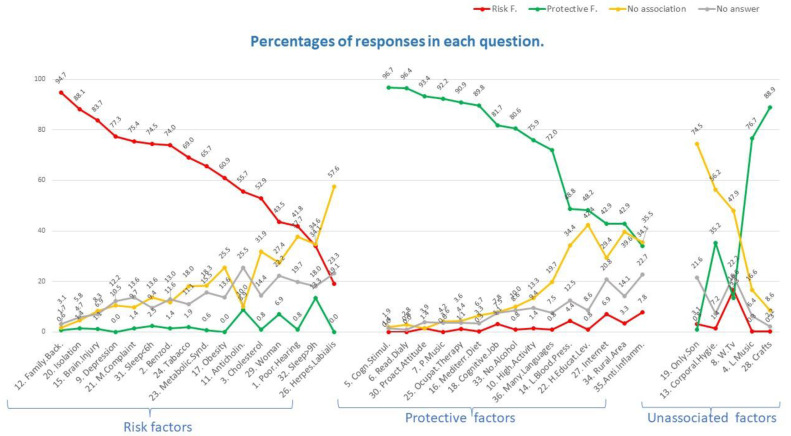
Percentages of responses that respondents gave for each question on the survey.

**Figure 2 ijerph-18-09934-f002:**
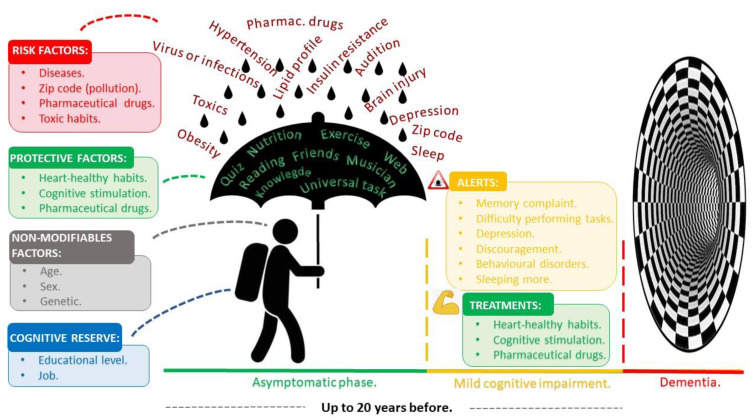
Phases of the evolution of dementia and the factors that influence the risk of dementia obtained from an updated bibliographic review (Table 2).

**Figure 3 ijerph-18-09934-f003:**
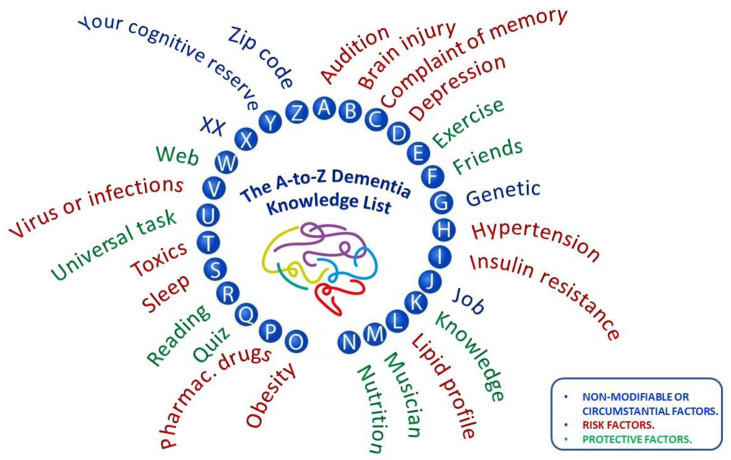
A-to-Z Dementia Knowledge List for screening and as a reminder tool.

**Table 1 ijerph-18-09934-t001:** Inclusion and exclusion criteria for the bibliographic review.

Inclusion Criteria	Exclusion Criteria
- Published in the last 3 years (2018–2021) - Published in PubMed or the Web of Science before March 2021 - Language: English - Key words: “cognitive impairment”, “dementia”, “subjective memory complaints”, “risk factors”, or “protective factors”	- Duplicated papers - No related papers - Screening using the title and abstract - Papers about molecular or non-commercialised drugs

**Table 2 ijerph-18-09934-t002:** A-to-Z Dementia Knowledge List.

Alphabet Letter	Dementia Factor	Reference
A	Audition	Livingston, 2020; WHO; ADI
B	Brain injury	Livingston, 2020, ADI
C	**Subjective memory Complaints**	Climent, 2018; Van Rooden, 2018; Borda, 2019; Numbers, 2020; Ramos, 2021.
D	Depression	Livingston, 2020; WHO; ADI
E	Exercise	Livingston, 2020; WHO; ADI
F	Friends	Livingston, 2020; WHO; ADI
G	Genetics	Cannon-Albright, 2019; Livingston, 2020; WHO; ADI; Pillai, 2021
H	Hypertension	Livingston, 2020; WHO; ADI; Pillai, 2021
I	Insulin resistance	Livingston, 2020; WHO; ADI
J	Job	Livingston, 2020
K	**Knowledge**	Rosenberg, 2018; Toman, 2018; Chen, 2020; Xie, 2020
L	Lipid profile	Svensson, 2019; Livingston, 2020; WHO; Iwagami, 2021; Kjeldsen, 2021; Pillai, 2021
M	Musician	Chan, 2018; Livingston, 2020
N	Nutrition	Livingston, 2020, WHO
O	Obesity	Livingston, 2020; WHO; ADI
P	**Pharmaceutical drugs**	Baek, 2019; Benito-León, 2019; Grossi, 2019; Newby, 2020; Weigand, 2020
Q	Quiz	Livingston, 2020, WHO
R	Reading	Livingston, 2020
S	Sleep	Ohara, 2018; Livingston, 2020
T	Toxics	Hahad, 2020; Livingston, 2020; WHO; ADI
U	Universal task	Livingston, 2020
V	**Viruses and infections**	Readhead, 2018; Tzeng, 2018; Warren-Gash, 2019; Woods, 2019; Abbott, 2020; Costa, 2021
W	**Web**	Krug, 2019; Neal 2021; Ramos 2021
X	XX-XY	ADI, Livingston, 2020, WHO
Y	Your cognitive reserve	Kremen, 2019; Livingston, 2020; WHO; ADI; Pillai, 2021
Z	Zip code	Peters 2019; Hahad, 2020; Livingston, 2020; ADI

The words in bold represent factors associated with dementia that are less well-known in the scientific literature.

**Table 3 ijerph-18-09934-t003:** Bibliographic citations related with new risk factors: subjective memory complaints, knowledge, pharmaceutical drugs, viral or other infections and use of the internet or digital technologies.

Factor	Study Type	Country (*N*)	Relationship to Dementia	Citation
Subjective memory complaint (SMC)	CI screening	Spain; *N* = 728	Sign of testing positive in CI screening.	Climent et al., 2018
	Image analysis	USA and The Netherlands; *N* = 25	Associated with greater white matter hyperintensity volume.	Van Rooden S, 2018
	Nation cohort	Mexico; *N* = 6327	Associated with an increase in the 3-year incidence of CI.	Borda et al., 2019
	CI screening and diagnosis	Australia; *N* = 873	Associated with a decline in global cognition over 6 years and may be predictive of the incidence of dementia.	Numbers et al., 2020
	CI screening and diagnosis	Spain; *N* = 281	Associated with a two-fold increase in detection of cognitive impairment, i.e., use of SMC as an inclusion criterium.	Ramos et al., 2021
Knowledge	Multicentre randomised controlled trial	Finland and Sweden; *N* = 1260	Multidomain lifestyle interventions have beneficial effects on cognition.	Rosenberg et al., 2018
	Review	Czech Republic	Multidomain lifestyle interventions generate significant effects in delaying cognitive decline.	Toman et al., 2018
	Interventional study	China; *N* = 1082	Patient adherence may be improved by increasing patients’ self-management efficacy.	Chen et al., 2020
	Randomised control trial	China; *N* = 148	Pragmatic multidomain interventions might supplement healthy aging policies if patients are empowered.	Xie et al., 2020
Pharmaceutical drugs	Retrospective	South Korea; *N* = 1,576,452	Association between benzodiazepine consumption and dementia after 5 years.	Baek et al., 2019
	Prospective	Spain; *N* = 5072	Association between a decreased risk of AD mortality and NSAID use. NSAIDs as a protective factor for developing AD.	Benito-León et al., 2019
	Retrospective	UK; *N* = 8216	ACB3 use is associated with dementia, especially in cognitively normal people. ACB12 and benzodiazepines are not associated with dementia.	Grossi et al., 2019
	Retrospective	UK, Spain, Denmark, and The Netherlands; *N* = 1127	Association between methotrexate use and lower dementia risk, especially when the therapy exceeds 4 years.	Newby et al., 2020
	Prospective	USA; *N* = 688	Anticholinergic drugs increase the risk of developing cognitive decline and dementia, especially among patients with genetic risk factors.	Weigand et al., 2020
Viral and bacterial infections	Molecular multiscale analysis	USA; *N* = 643 brain samples	Increased HHV-6A and HHV-7 in the brains of patients with AD.	Readhead et al., 2018
	Retrospective	Taiwan; *N* = 33,448 participants	Patients with HSV infections may have a 2.56-fold increased risk of developing dementia. Treatment with antiherpetic medications was associated with a decreased risk of dementia.	Tzeng et al., 2018
	Systematic review	England; *N* = 57 studies	Recent reactivation of herpesvirus HSV-1, HSV-2, cytomegalovirus, and HHV-6 may be associated with dementia or MCIs.	Warren–Gash et al., 2019
	Review	Australia; *N* = 15 studies	*Chlamydia pneumoniae* found at increased rates in the brains of AD patients. Limitations of previous human and animal studies preclude conclusive interpretation.	Woods et al., 2019
	Systematic review	Brazil; *N* = 9 studies	Infection by *Porphyromonas gingivalis* and bacterial lipopolysaccharide administration appears to be related to the pathogenesis of AD.	Costa et al., 2021
Use of the internet or digital technologies	Longitudinal study	Brazil; *N* = 2902	Older adults who continue using the internet were more likely to have significantly lower cognitive loss.	Krug et al., 2019
	Case–control study	Spain; *N* = 497 participants	Internet use was a factor associated with a 67–86% reduced risk of CI compatible scores in neuropsychological tests.	Ramos et al., 2020
	Systematic review	The Netherlands *N* = 9 studies	Weak evidence that digital technologies may provide less benefit to people with dementia than those with an MCI.	Neal et al., 2021

CI: cognitive impairment; ACB: anticholinergic burden scale; AD: Alzheimer disease; HVS: Herpes simplex virus; HSV-1: Herpes simplex virus type 1; HSV-2: Herpes simplex virus type 2; HHV-6: Human herpesvirus 6; HHV-7: Human herpesvirus 7; SMC: subjective memory complaint; MCI: mild cognitive impairment; NSAIDs: non-steroidal anti-inflammatory drugs; USA: United States of America; UK: United Kingdom.

**Table 4 ijerph-18-09934-t004:** Summary of the respondent profiles and mean number of correct items.

Respondent Profile Description		Risk Factors (16)	Protective Factors (15)	Non-Associated Factors (5)
	*n* (%)	mean ± *SD*	mean ± *SD*	mean ± *SD*
Age				
[20–30]	45 (12.5)	11.2 ± 3.4 a	11.6 ± 2.5 a	1.7 ± 1.1 a
[30–45]	99 (27.4)	10.5 ± 3.4 ab	10.5 ± 2.8 a	1.9 ± 1.3 a
[45–60]	140 (38.8)	9.9 ± 3.8 ab	11.0 ± 2.6 a	2.1 ± 1.2 a
[60–75]	77 (21.3)	9.2 ± 3.7 b*	10.6 ± 2.8 a	2.2 ± 1.4 a
Years of experience				
[0–10]	75 (20.9)	11.4 ± 3.3 a	11.4 ± 2.6 a	1.8 ± 1.2 a
[10–20]	81 (22.6)	9.8 ± 3.4 b*	10.2 ± 2.9 b*	2.0 ± 1.2 a
[20–30]	102 (28.4)	9.9 ± 3.8 b*	11.0 ± 2.7 ab	2.1 ± 1.2 a
[30–40]	72 (20.1)	10.0 ± 3.7 ab	10.9 ± 2.2 ab	2.2 ± 1.3 a
[40–50]	29 (8.1)	8.6 ± 3.8 b**	10.7 ± 3.2 ab	2.3 ± 1.4 a
Close relatives with dementia				
Yes	248 (68.7)	10.2 ± 3.5 c	10.9 ± 2.6 d	2.0 ± 1.3 e
No	113 (31.3)	9.9 ± 4.0 c	10.8 ± 2.9 d	2.1 ± 1.2 e
Total	361 (100)	10.1 ± 3.7	10.9 ± 2.7	2.0 ± 1.3

Description of the respondent profiles and mean number of correct items together with their standard deviations. Tukey multiple comparison tests (a) significantly higher mean number of correct items; (b) significantly lower mean number of correct items; (ab) mean number of correct items not significantly different from the mean number of correct items a and b; *t*-test for independent samples; (c, d and e) mean number of correct elements not significantly different between the Yes and No group; * *p*-value < 0.05; ** *p*-value < 0.01 with respect to group a.

**Table 5 ijerph-18-09934-t005:** Survey questions on participant knowledge of the factors associated with the development of dementia and the distribution of the responses from respondents (correct answers are shown in bold).

Questionnaire Questions	Risk Factor *n* (%)	Protective Factor *n* (%)	Non-Associated Factor *n* (%)	No Answer *n* (%)
1. Hearing loss is:	**151 (41.8)**	3 (0.8)	136 (37.7)	71 (19.7)
2. Chronic benzodiazepine use is:	**267 (74.0)**	5 (1.4)	42 (11.6)	47 (13)
3. Elevated levels of total cholesterol constitute a:	**191 (52.9)**	3 (0.8)	115 (31.9)	52 (14.4)
4. Listening to music daily is:	1 (0.3)	277 (76.7)	**60 (16.6)**	23 (6.4)
5. Cognitive stimulation constitutes a:	0 (0.0)	**349 (96.7)**	7 (1.9)	5 (1.4)
6. Reading daily constitutes a:	0 (0.0)	**348 (96.4)**	10 (2.8)	3 (0.8)
7. Playing a musical instrument is usually:	0 (0.0)	**333 (92.3)**	15 (4.2)	13 (3.6)
8. Watching television daily is:	60 (16.6)	48 (13.3)	**173 (47.9)**	80 (22.2)
9. Suffering from depression years before (more than 10) the development of dementia is a:	**279 (77.3)**	0 (0.0)	38 (10.5)	44 (12.2)
10. High levels of physical activity constitute a:	5 (1.4)	**274 (75.9)**	48 (13.3)	34 (9.4)
11. The use of anticholinergic drugs constitutes a:	**201 (55.7)**	32 (8.9)	36 (10.0)	92 (25.5)
12. Having a family history of some type of dementia is:	**342 (94.7)**	2 (0.6)	6 (1.7)	11 (3.1)
13. Maintaining body hygiene and cleanliness is:	5 (1.4)	127 (35.2)	**203 (56.2)**	26 (7.2)
14. Controlling systolic blood pressure in middle-aged people is:	16 (4.4)	**176 (48.8)**	124 (34.4)	45 (12.5)
15. Brain trauma constitutes a:	**302 (83.7)**	4 (1.1)	30 (8.3)	25 (6.9)
16. A high adherence to the Mediterranean diet is a:	1 (0.3)	**324 (89.8)**	24 (6.7)	12 (3.3)
17. Obesity, understood as a body mass index greater than 30 kg/m^2^, is a:	**220 (60.9)**	0 (0.0)	92 (25.5)	49 (13.6)
18. Jobs with a higher cognitive level constitute a:	11 (3.1)	**295 (81.7)**	28 (7.8)	27 (7.5)
19. To be an only child is:	11 (3.1)	3 (0.8)	**269 (74.5)**	78 (21.6)
20. Social isolation is a:	**318 (88.1)**	5 (1.4)	17 (4.7)	21 (5.8)
21. Memory complaints constitute a:	**272 (75.4)**	5 (1.4)	35 (9.7)	49 (13.6)
22. A high level of education constitutes a:	3 (0.8)	**174 (48.2)**	153 (42.4)	31 (8.6)
23. Metabolic syndrome is a:	**237 (65.7)**	2 (0.6)	66 (18.3)	56 (15.5)
24. Tobacco use constitutes a:	**249 (69.0)**	7 (1.9)	65 (18.0)	40 (11.1)
25. Occupational therapy in the elderly population is a:	4 (1.1)	**328 (90.9)**	16 (4.4)	13 (3.6)
26. Being prone to cold sores is:	**69 (19.1)**	0 (0.0)	208 (57.6)	84 (23.3)
27. Using the internet or social networks from any technological device is:	25 (6.9)	**155 (42.9)**	106 (29.4)	75 (20.8)
28. Manual work is:	1 (0.3)	321 (88.9)	**31 (8.6)**	8 (2.2)
29. Being a woman constitutes a:	**157 (43.5)**	25 (6.9)	99 (27.4)	80 (22.2)
30. That people acquire a proactive attitude of prevention regarding the modifiable factors associated with dementia is:	5 (1.4)	**337 (93.4)**	5 (1.4)	14 (3.9)
31. Sleeping less than 6 h a day:	**269 (74.5)**	9 (2.5)	49 (13.6)	34 (9.4)
32. Sleeping more than 9 h a day:	**123 (34.1)**	488 (13.3)	125 (34.6)	65 (18.0)
33. Refraining from drinking alcohol is:	3 (0.8)	**291 (80.6)**	36 (10.0)	31 (8.6)
34. Living in the countryside or in rural areas is:	12 (3.3)	**155 (42.9)**	143 (39.6)	51 (14.2)
35. Taking anti-inflammatory drugs and/or having inflammatory diseases under control is:	28 (7.8)	**123 (34.1)**	128 (35.5)	82 (22.7)
36. Speaking several languages is:	3 (0.8)	**260 (72.1)**	71 (16.7)	27 (7.5)

## Data Availability

This data used for this study is available upon request.

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
