# Peer review of "Pharmacists’ Knowledge of Factors Associated with Dementia: The A-to-Z Dementia Knowledge List"

_ijerph, 2021, doi:10.3390/ijerph18199934_

Round 1

Reviewer 1 Report

An interesting paper, highlighting the importance of Continued Professional Development (CPD) in health/allied health professions.

There are a few minor language edits/clarifications required (see list below).

With regards to the discussion of the content, this reviewer finds some of the commentary to be biased in particular directions, without further scientific investigation or explanation.  For example, the generally accepted notion that cholesterol is a risk factor - there is evidence that this depends on genetic links (APOE 4 status) (see Pillai et al, Journal of Alzheimers Disease 2021).  Also while attention is importantly drawn to viral infections as risk factors, there is very little on pollutants, for example mycotoxins. 

This reviewer finds the focus of this paper lacking, in is this a paper highlighting lack of scientific knowledge of the survey participants, and an effort to remedy those?  Or is the focus on CPD/ Educational interventions for health care workers?
Depending on which, I would suggest revision to clarify the focus - scientific or education (or indeed publish two separate papers on each area).

Edits:

Line 16: Punctuation - ...no cure, that can begin
Line 19: Proximity to what?
Line 23: of 361, instead of in 361
Line 77: In recent years (instead of in the recent)
Line 197: Clear evidence of what?  Please clarify
Line 208: Viral infections, instead of virus infections (also Line 328)
Line 212: Alzheimer's Disease
Line 323: The word backpack was originally used and then changes here to bag (also Line 467).  For consistency, use either backpack or bag throughout

Reviewer 2 Report

Some major concerns are: 

1. Some results are mixed in the Methods and Materials section.

2. Results and some discussions are mixed. 

2. The A-to-Z Dementia Knowledge List was constructed mostly on Livingston, 2020; WHO; ADI. More comprehensive references are needed for establishing such an important dementia knowledge list 

3. One conclusion from this study is that "younger participants had a better knowledge of the factors associated with dementia". What is the responding % of the younger group in the participants? This number can bias the conclusion.

4. For the association tests, it is recommend to add interactions, i.e., explaining possible interactions between factors.

5. Any reasoning about low responding rate (7.4%) in collection data? Answers may affect how to interpret the results of this study.

Round 2

Reviewer 2 Report

1. In the "Statistical analysis" section, it says, "Based on the classical statistical formula for the calculation of the required sample size in a study on the estimation of a mean score, with 95% confidence, in a questionnaire with an estimated standard deviation of 4 points and a precision of 1 point, 246 participants were needed. According to our estimations, a 5% of response rate was needed to achieve our objective." Please add the necessary formula to justify the numbers better here.

2. Something went wrong in the revision in the Results section. For example, the result context for Table 3 is entirely missing. 

3. Unaddressed in version 2: 
1) The reasoning explaining "the younger participants had a better knowledge of the factors associated with dementia": at least from Table 4, this "younger age" conclusion does not hold; and 
2) it is suggested in version 1's improvement to add interaction analysis, for example, interactions among the age, experience years, and close relatives with dementia. But the interactions were not taken into account in the revision.
